# *CRB1*-Related Retinal Dystrophies in a Cohort of 50 Patients: A Reappraisal in the Light of Specific Müller Cell and Photoreceptor *CRB1* Isoforms

**DOI:** 10.3390/ijms222312642

**Published:** 2021-11-23

**Authors:** Kévin Mairot, Vasily Smirnov, Béatrice Bocquet, Gilles Labesse, Carl Arndt, Sabine Defoort-Dhellemmes, Xavier Zanlonghi, Dalil Hamroun, Danièle Denis, Marie-Christine Picot, Thierry David, Olivier Grunewald, Mako Pégart, Hélèna Huguet, Anne-Françoise Roux, Vasiliki Kalatzis, Claire-Marie Dhaenens, Isabelle Meunier

**Affiliations:** 1Department of Ophthalmology, University North Hospital of Marseille, Sensgene Care Network, 13915 Marseille, France; kevinmairot@gmail.com (K.M.); daniele.denis@ap-hm.fr (D.D.); thierry.david@ap-hm.fr (T.D.); 2Department of Visual Exploration and Neuro-Ophthalmology, Robert Salengro Hospital, Sensgene Care Network, 59045 Lille, France; vasily.smirnov@chru-lille.fr (V.S.); sabine.defoort@chu-lille.fr (S.D.-D.); 3National Reference Centre for Inherited Sensory Diseases, University of Montpellier, Montpellier University Hospital, Sensgene Care Network, ERN-EYE Network, 34000 Montpellier, France; beatrice.bocquet@inserm.fr (B.B.); mako.pegart@chu-montpellier.fr (M.P.); 4Institute for Neurosciences of Montpellier (INM), University of Montpellier, INSERM, 34000 Montpellier, France; anne-francoise.roux@inserm.fr (A.-F.R.); vasiliki.kalatzis@inserm.fr (V.K.); 5Structural Biochemistry Centre, University of Montpellier, INSERM, CNRS, 34000 Montpellier, France; labesse@cbs.cnrs.fr; 6Department of Ophthalmology, Reims University Hospital, 51000 Reims, France; carndt@chu-reims.fr; 7Department of Ophthalmology, Rennes University Hospital, 35000 Rennes, France; dr.zanlonghi@gmail.com; 8Department of Research and Innovation, University of Montpellier, Montpellier University Hospital, 34000 Montpellier, France; d-hamroun@chu-montpellier.fr; 9Clinical Investigation Center (CIC), Clinical Research and Epidemiology Unit (URCE), University of Montpellier, 34000 Montpellier, France; mc-picot@chu-montpellier.fr (M.-C.P.); h-huguet@chu-montpellier.fr (H.H.); 10Inserm, Lille University Hospital, U1172-LilNCog-Lille Neuroscience and Cognition, University of Lille, 59045 Lille, France; olivier.grunewald@chru-lille.fr (O.G.); claire-marie.dhaenens@inserm.fr (C.-M.D.); 11Molecular Genetics Laboratory, University of Montpellier, Montpellier University Hospital, 34000 Montpellier, France

**Keywords:** *CRB1*, isoforms, early onset retinal dystrophy, Leber congenital amaurosis, macular dystrophy, Müller cells, pathogenic variant, photoreceptors, rod-cone dystrophy, spectral domain optical coherence tomography

## Abstract

Pathogenic variants in *CRB1* lead to diverse recessive retinal disorders from severe Leber congenital amaurosis to isolated macular dystrophy. Until recently, no clear phenotype-genotype correlation and no appropriate mouse models existed. Herein, we reappraise the phenotype-genotype correlation of 50 patients with regards to the recently identified *CRB1* isoforms: a canonical long isoform A localized in Müller cells (12 exons) and a short isoform B predominant in photoreceptors (7 exons). Twenty-eight patients with early onset retinal dystrophy (EORD) consistently had a severe Müller impairment, with variable impact on the photoreceptors, regardless of isoform B expression. Among them, two patients expressing wild type isoform B carried one variant in exon 12, which specifically damaged intracellular protein interactions in Müller cells. Thirteen retinitis pigmentosa patients had mainly missense variants in laminin G-like domains and expressed at least 50% of isoform A. Eight patients with the c.498_506del variant had macular dystrophy. In one family homozygous for the c.1562C>T variant, the brother had EORD and the sister macular dystrophy. In contrast with the mouse model, these data highlight the key role of Müller cells in the severity of *CRB1*-related dystrophies in humans, which should be taken into consideration for future clinical trials.

## 1. Introduction

Inherited retinal dystrophies (IRDs), despite clinical and genetic heterogeneity, share progressive degeneration of photoreceptors and/or retinal pigment epithelium cells. IRDs affect about 2 million people worldwide among the 300 million patients with inherited diseases. They account for 10% of the causes of visual impairment in Western countries. IRDs are classified according to the predominant cell type affected (rod-cone dystrophy, RCD, versus cone-rod dystrophy, CRD) and/or topography of the lesions (maculopathy versus generalized retinal dystrophy). Thus, full-field electroretinogram (ff-ERG) and multimodal imaging are both essential for IRD classification. RCD (also known as retinitis pigmentosa, RP) is the most frequent phenotype with an estimated prevalence of 1/4000. Maculopathies account for 20% of cases, with Stargardt disease being the most prevalent [1,2]. A distinction is made between IRDs with an early onset (early onset retinal dystrophy, EORD), including Leber congenital amaurosis (LCA), and those with a later onset after the age of 5 years. All modes of inheritance have been reported in these disorders, autosomal recessive (aR), dominant (aD), X-linked or, more rarely, mitochondrial.

Since the advent of next-generation sequencing techniques, LCA-related genes, such as *CRX* or *RDH12,* have been unexpectedly linked to isolated maculopathies [3,4,5,6]. Tsang et al. reported the first cases of macular dystrophy with bi-allelic pathogenic variants in *CRB1* [7]. Prior to this, *CRB1* had only been associated with EORD (where it was responsible for 10 to 13% of LCA cases) or RP (responsible for 8% of cases) with specific clinical features, such as para-arteriolar preservation of the retinal pigment epithelium, peripheral nummular pigmentation or Coats-like disease [8,9,10,11,12,13,14,15,16].

The *CRB1* (Crumbs homologue 1) protein belongs to the CRB complex, an apical complex important for cell junctions, and acts as a cell polarity regulator (see [17] for review); it also plays a major role during retinal development [13,18,19,20]. The *CRB1* gene contains 12 exons spanning 210 kb with 12 predicted transcripts using alternative exons (https://www.ensembl.org (accessed on 25 October 2021)). In the retina, two major transcripts, which encode cell-surface proteins, are mainly expressed in Müller cells and in photoreceptors (Figure 1 and Figure 2) [21]. Müller cells express exclusively the canonical transcript named *CRB1-A* [MT470365] encoding a 1406 amino acid protein with 19 EGF-like domains and 3 laminin G domains. Photoreceptors mainly express the *CRB1-B* [MT470366] transcript encoding a 1003 amino acid protein. Transcript A is the most abundant during development and transcript B is predominant in adult retina [21]. A third transcript named *CRB1-C* [MT470367] encoding a 754 amino acid protein without transmembrane and intracellular domains was also reported but is less abundant than the *CRB1-A* and *CRB1-B* transcripts [21]. To date, 457 *CRB1* pathogenic variations have been described in HGMD professional 2021.3 database (https://hgmd/pro/ (accessed on 25 October 2021)): 333 missense/nonsense variants, 28 splice variants, 1 regulatory substitution, 86 small insertions/deletions and 9 large insertions/deletions. One specific pathogenic variation, c.498-506del, which is an in-frame deletion, is specifically encountered in *CRB1* related macular phenotypes [7,22].

In this retrospective multicenter observational study, we reappraise for the first time our cohort of 50 patients with bi-allelic pathogenic variants in *CRB1* to refine phenotype-genotype correlations in light of recent knowledge on *CRB1* isoforms and their retinal cell expression.

## 2. Results

### 2.1. Clinical Data

Fifty patients (45 families) with bi-allelic pathogenic variants were included in the study. The group comprised 25 men and 25 women. Thirty-seven patients were reexamined with a median follow-up of 3.9 years (range 3 months to 19 years). Clinical and imaging data are summarized in Table 1. Twenty-eight patients were classified as EORD, thirteen as RP and nine as isolated macular dystrophy based on age of onset, symptomatology, multimodal imaging and ff-ERG responses. In one family (M-1640), the brother had EORD and the sister macular dystrophy (Figure 3).

Considering the classical anomalies encountered in *CRB1* cases, twelve patients had papillary drusen and fourteen had para-arteriolar preservation of the retinal pigment epithelium. In generalized retinal dystrophy with exclusion of LCA cases (no relevant OCT measurements), we observed a macular thickening with loss of the retinal reflectivity segmentation pattern. This thickening was noted during the first three decades compared to a control group (Table 2). Such a retinal thickening is not the rule in retinal dystrophies, which are characterized by a constant and progressive cell loss. This is related to the specific role of *CRB1* during retinal development, thus, the retina is initially thickened and disorganized. Macular cystoid edema occurred mainly in the first two decades in the EORD or RP patients, and in the third decade in maculopathy patients (Figure 4). In the macular group, visual anomalies were noted during the third decade (mean age of onset: 23.5; range 16 to 42 years). In the family of proband L-93112991 with initial maculopathy, the proband progressed with time to a generalized retinal dystrophy whereas his brother had a less severe progression.

### 2.2. Genetic Results

All genetic data are summarized in Table 3. In our cohort, 41 different pathogenic variants are reported: 26 missense (63% of the total variants), nine indel, four nonsense variants, one splice variation and one leading to the loss of the final stop codon, resulting in a protein extended by 110 amino acids. Most of these variants (27/41) are localized in the exons 6, 7 and 9, thus affecting both *CRB1-A* and *CRB1-B* isoforms (Figure 2 and Figure 5). 

Bi-allelic indel or nonsense pathogenic variants leading to loss of function, due to nonsense mediated decay (NMD) activation, were noted in seven EORD cases versus none in RP cases (Table 3). Bi-allelic missense pathogenic variants were observed in 14 EORD and in nine RP cases. The recurrent pathogenic variant c.2290C>T, p.(Arg764Cys) [23] was frequent and carried by ten patients. Four of them were homozygous for this variation, three presenting EORD and one RP. All patients with a macular phenotype, except one, carried one allele with the c.498_506del, p.(Ile167_Gly169del) variant. The remaining patient with a macular phenotype was homozygous for the c.1562C>T, p.(Ala521Val) variant, whereas his brother had an EORD.

### 2.3. Assessment of the Functional Effect of CRB1 Variants

We identified a number of known pathogenic *CRB1* variants in our cohort: p.Gly169Valfs*3 [24]; p.Ile205Aspfs*13 [25]; c.653-1G > T [26]; p.Gln362* [27]; p.Thr745Met [23]; p.Thr745Lys [28]; p.Arg764Cys [23]; p.Gly770Ser [29]; p.Lys801* [30]; p.Gly827* [31]; p.Pro836Thr and p.Gly850Ser [9]; p.Asn880Ser [32]; p.Cys948Tyr [23]; p.Asp1005Val [16]; p.Ile1100Thr [33]; p.Gly1103Arg [34]; p.Gly1103Val [26]; p.Leu1107Pro [31]; p.Tyr1161Cys [35]; p.cys1229Ser [36]; p.Ile1358Asn [37]; p.Pro1381Arg [38]. 

In addition, we identified 17 novel variants (Table 4). Among these, three indels and the nonsense variant in exon 3 probably lead to NMD, as they are situated before the penultimate exon. This is not the case for the two novel variants in exon 12 that presumably lead to an extension of the intracellular part of the protein, a region interacting with other partners, such as PALS1 (protein associated with lin seven 1) [17,39,40,41], which stabilizes the CRB complex, and Moesin [17,42], which combines with the FERM-binding domain and is involved in epithelial to mesenchymal transition. We also describe nine novel missense variations. Two of these variants, p.(Ala521Val) and p.(Cys606Ser), are predicted to destabilize the laminin domain by bringing a larger hydrophobic residue valine into a buried position or by breaking a conserved disulfide bridge (Figure 6A). A similar situation is encountered for two variants, p.(Gly850Arg) and p.(Leu862Arg), in the second laminin domain (Figure 6B). The large and polar arginine residues can hardly be accommodated in the buried and compact environment of Gly850 and Leu862. By contrast, the variant p.(Tyr808Cys) is predicted to occur at the surface of this domain. However, the added cysteine would favor improper disulfide bridge formation in the extracellular environment, resulting in incorrect macromolecular assembly. The variant p.(Val1055Glu) again brings a larger and polar residue into a position buried in the third laminin domain, and hence, it is also predicted to strongly destabilize the corresponding domain (Figure 6C). Finally, the variant p.(Cys1238Phe) will abolish a predicted disulfide bridge in another EGF-like domain and again prevent its proper folding or impact its overall stability (Figure 6F). 

### 2.4. Impact of CRB1-A and CRB1-B Variants on the Phenotype

We then reappraised the clinical data regarding the localization and the consequences of the genomic variants on both isoforms: the specific Müller isoform *CRB1-A* (encoded by 12 exons) and the photoreceptor isoform *CRB1-B* (7 exons from 5c to 11). The two isoforms differ in their N- and C-termini (Figure 2). The predicated impact of the different pathogenic variants on the isoforms, in terms of the estimated levels of mutant and wild type isoforms in Müller cells and in photoreceptors, is described in Table 3. The *CRB1* isoform B has no impact on the severity of the phenotype as EORD, RP and macular dystrophy can occur regardless of the level of isoform B (Figure 7). For example, patients M-116 and M-4621 presenting an EORD, carried two pathogenic variants (c.4219T>A and c.4142C>G, respectively) that did not affect isoform B. These variants are localized in exon 12 and affect the cytoplasmic C-terminal domains. The patient M-4621 was homozygous for the pathogenic variant c.4142C>G, p.(Pro1381Arg) which modifies the FERM-binding domain interacting with the protein membrane Moesin (Figure 8). In patient M-116, no isoform A was produced, as both alleles carried loss-of-function variants affecting specifically the Müller cell isoform. The p.(Trp229*) variant in exon 3 is predicted to lead to a shorter *CRB1-A* protein truncated after the 5th EGF-like domain, thus, if the protein were expressed, it would be likely degraded. On the other allele, the p.(Ter1407Lysext*111) variant leads to an abnormal C-terminal elongated isoform, affecting the PDZ domain. The second example of a dystrophy associated with the normal isoform B is the maculopathy presented by the two patients (M-2183 and M-3343) carrying the specific c.498_506del macular variant. Both patients carried on the second allele variants located in exons 4 and 12, respectively, that did not affect isoform B. Patient M-3343 also carried the pathogenic variant p.(Pro1381Arg) modifying the FERM-binding domain. Furthermore, all patients with the specific c.498_506del macular variant developed a macular dystrophy regardless of the impact on isoform B. 

Among the 13 patients with RP, nine had missense variants on both alleles and four had one missense and one frameshift variant. None of the RP patients presented a pathogenic variant in the intracytoplasmic domain of the isoform *CRB1-A* nor in the EGF-like 8 domain encoded by exon 4. 

Interestingly, the p.(Lys801*) variant, which leads to either a shorter protein or to the absence of protein synthesis via NMD activation, was associated with two different phenotypes when present in *trans* with a variant specific to the Müller cell isoform. Patients M-3073-1 and 3073-3 carrying p.(Lys801*) in *trans* with the c.498_506del variant affecting the EGF-like 4 domain (exon 2) manifested a maculopathy, whereas patient M-1732 carrying p.(Lys801*) in *trans* with c.974G>T, p.(Cys325Phe) had EORD. This latter variant disrupts a disulfide bond in the EGF-like 8 domain. Another cysteine change in the EGF-like 8 domain of exon 4 in position 316 was observed in patient M-2183, also a carrier in *trans* of c.498_506del. The patient manifested a maculopathy with an abnormal ERG showing reduced (50%) rod and cone responses.

## 3. Discussion

We demonstrate that *CRB1*-related maculopathy is not a rare phenotype. In this large cohort, it occurs in nine patients, of which all except one carried the specific c.498_506del pathogenic variant in one allele. Bi-allelic nonsense and frameshift variants can be noted in EORD, including LCA, as well as in RP, but NMD is more frequently predicted in EORD (22/28) than in RP (2/13). In this cohort, macular dystrophy and EORD occurs even if the *CRB1-B* isoform is not impacted by pathogenic *CRB1* variants. 

### 3.1. CRB1-Related Retinal Dystrophies

All the patients studied necessarily had bi-allelic variants modifying *CRB1-A*, the canonical long isoform, as the next generation sequencing (NGS) gene panel used for molecular diagnosis has not been designed to screen the specific region of the shorter isoform B. These two isoforms have a specific cell distribution and a differential expression pattern in the retina during the embryonic and adult periods [21]. *CRB1-A* is localized to the apical tips of Müller cells and *CRB1-B* in the inner and outer segments with a gradient (Figure 1). These cell-surface proteins are both present at the level of the outer limiting membrane (OLM), which is formed by the adherens junctions between the adjacent Müller cells and photoreceptors, creating a distinct barrier between the neural retina and the inner/outer segments. *CRB1* expression is variable: *CRB1-A* is the predominant isoform during developmental stages. By contrast, *CRB1-B* is by far the most abundant isoform in the adult human, as well as murine retina. Thus, there is a relative *CRB1-A* depletion in mature human retina.

#### 3.1.1. The Macular Phenotype Is Determined by the c.498_506del Variant Specific to *CRB1-A*

All patients with macular dystrophy, except one, had the variant c.498_506del specific to isoform A. It appears that a maculopathy can also be observed even if isoform B is normal, such as in patients M-2183 and M-3343. The remaining patients with a macular dystrophy carried c.498_506del with in *trans* missense and nonsense variants also found in EORD. The maculopathy is not the consequence of the impairment of *CRB1* at developmental stages, as the lamination of the retina is preserved outside the macular zone on SD-OCT and its onset occurs most frequently in the third decade of life. 

The specific macular c.498_506del variant leads to the formation of a protein lacking 3 amino acids belonging to the Ca^2+^ EGF-like 4 domain. With the lack of a corresponding mouse model, the mechanism of this variant is not clearly known. Recently, human induced pluripotent stem cells [43] were generated from a patient with a macular dystrophy carrying the c.498_506del in *trans* with the c.613_619del variant in the aim of studying the dysfunction and changes in Müller cells and photoreceptors in human retinal organoids. These patient-specific organoids will provide valuable insights into the specific macular pathophysiology specifically associated with the c.498_506del variant. We can hypothesize that this in-frame deletion abolishes a calcium binding site in two consecutive EGF-like modules (EGF-4 and EGF-5) of *CRB1*. The deletion is also expected to strongly destabilize the local folding, as well as the relative orientation of those two EGF-like modules, and hence the proper structural organization of the whole protein. 

Despite the lack of a precise mechanism, our data underline that the variant c.498_506del drives the macular phenotype independently of *CRB1-B* photoreceptor impairment. 

#### 3.1.2. *CRB1* Variants and Structure Function Correlation in EORD and in RP 

Twenty-six patients with EORD had a combined impairment of one or two alleles of the *CRB1-B* isoform. A haploinsufficiency in *CRB1-B*, the photoreceptor isoform, is not the mechanism that leads to a generalized retinal phenotype, as patients M-3073-1 and -3 who had 50% of the WT *CRB1-B* isoform only presented with a macular dystrophy. Moreover, it should be noted that two patients with bi-allelic variants only belonging to isoform A, and thus producing a normal photoreceptor isoform B, still presented with EORD. 

Taken together, these data suggest that the phenotype in *CRB1* patients is also dependent on the severity of the Müller cell impairment related to specific *CRB1-A* variant combinations. The two patients with EORD and WT *CRB1-B* both have variants in exon 12. One patient is homozygous for the variant c.4142C>G p.(Pro1381Arg) in exon 12. The second patient is pseudo-homozygous for the variant c.4219T>A p.(Ter1407Lysext*111) in exon 12 with an abnormal elongation of the protein in its intracellular part, as the variant in *trans*, c.687G>A (p.Trp229*) in exon 3, leads to NMD. It appears that there is a high severity associated with variants modifying the transmembrane or intracellular domains (exon 12). 

Mutations in exon 12 likely affect the functioning of *CRB1* in different manners. First, p.(Ile1358Asn) is predicted to destabilize the transmembrane segment due to the replacement of a hydrophobic isoleucine with a polar asparagine. This could lead to improper membrane insertion or alternatively to improper multimerization to hide the polar sidechain. The variant p.(Pro1381Arg) occurs in the vicinity of the phosphorylation site Thr1378 (see https://www.phosphosite.org/ (accessed on 25 October 2021)). The positively charged arginine could either (i) prevent recognition by the protein-kinase or (ii) stabilize the phosphorylated form of *CRB1* and hence prevent association with other partners, such as PALS1 and other PDZ-containing proteins. The two other mutants have lost their PDZ binding motif in the C-terminus of *CRB1* due to sequence extension and/or substitution/addition of a charged residue (arginine or lysine). Such variants would directly prevent interactions with various PDZ-containing partners, such as PALS1. This likely impacts the function of *CRB1* to the same extent as the two other mutations discussed above. 

Among the patients with the c.2401A>T, p.(Lys801*) variant and 50% WT isoform B (M-1732, M-3073-1 and M-3073-3), patient M-1732 who carries in *trans* the variant c.974G>T, p.(Cys325Phe) had EORD. In the same manner, patient M-2183 with the c.946T>C, p.(Cys316Arg) had a moderate decrease in rod and cone responses. These cases could suggest that variants affecting the EGF-like 8 domain in exon 4 could drive a more severe Müller cell impairment. Regarding the two mutations in exon 4, p.(Cys325Phe) and p.(Cys316Arg), modeling of the corresponding EGF-like domain suggests that both would be deleterious for the proper folding and/or stability of the local structure: p.(Cys325Phe) would prevent the formation of a predicted disulfide bridge important for the EGF-like structure; p.(Cys316Arg) would replace a small and buried residue with a large and polar arginine, likely preventing proper folding of the corresponding domain. It is also predicted to break up a conserved disulfide bridge, similar to p.(Cys325Phe).

Patients with RP have mainly missense variants and none of them carry a pathogenic variant in *CRB1* exons 4 or 12. This again highlights that the EGF-like 8 and the intracellular domains are essential for the proper function of Müller cells. 

### 3.2. Intrafamilial Variation Is an Exception 

Among the 45 families studied, a heterogeneity in phenotype was only noted in one consanguineous family. In family M-1640, homozygous for the novel c.1562C>T, p.(Ala521Val) variant in exon 6, the proband had EORD with indiscernible responses of ff-ERG, whereas his sister had a maculopathy with discernible responses. There was no additional pathogenic variant in any of the 230 retinal dystrophy genes analyzed on this NGS panel, which included the *CRB2* gene. As the c.1562C>T variant is located in an exon shared by both isoforms A and B, we cannot rule out that the phenotypic difference could be explained by an additional variant present in the *CRB1* isoform B-specific regions not covered by the NGS panel (N-terminus and C-terminal parts including 58 amino acids). Lastly, a splice defect of this variant was ruled out according to MobiDetails prediction. 

### 3.3. Limitations of Mouse Models, Concordance and Discordance 

Diverse *Crb1* mouse models have been reported to date [17]. Recently, Ray et al. designed specific novel mutant mouse models to evaluate each isoform separately (*Crb1^ex1^*, *Crb1^delB/delB^*), as well as their combined functions (*Crb1^null^*) [21]. All the *CRB1* transcripts are effectively disrupted in the novel *Crb1^null^* mouse model (CRISPR-mediated deletion of exons common to all *Crb1* isoforms) [21]. Similarly, *Crb1-A* and *Crb1-B* are both disrupted in previously described rd8 mouse models (1-bp frameshift deletion in exon 9 leading to the loss of the transmembrane and intracellular domains), whereas *Crb1-A* isoform is specifically lost in the *Crb1^ex1^* knockout model (deletion of the promotor region of the first exon of canonical isoform A) [17]. By contrast, isoform B is specifically lost in the novel designed *Crb1^delB/null^* and in *Crb1^delB/delB^* models (CRISPR-mediated deletion in the promotor and the first exon of isoform B) [21]. Photoreceptor displacement and rosette formation were described in rd8 and *Crb1^ex1^* knockout model, due to the loss of adhesion between photoreceptors and Müller cells [17]. Ray et al. [21] reported similar OLM gaps with loss of adherens junctions between photoreceptors and Müller cells, and with an abnormal shift of the photoreceptor nucleus to its inner segment in *Crb1^null^* (all retinal isoforms A, B and C are disrupted), as well as in *Crb1^delB/null^* and in *Crb1^delB/delB^* models (CRISPR-mediated deletion in the promotor and the first exon of isoform B) but with a lower frequency [21]. Structural alterations in number and size of the apical microvilli of Müller cells were also described in rd8 and knockout models [17]. All these models indicate that the lack of one or both isoforms induces OLM changes with loss of photoreceptor-Müller cell contacts combined with structural changes in both cell types. 

Beyond Müller cell-photoreceptor adhesion and interaction, photoreceptor loss was not significant in mice lacking only one of the two isoforms, except in aged mice. A retinal degeneration with photoreceptor loss occurs only in *Crb1^null^* mice lacking all *CRB1* isoforms. 

Based on the characterization of the previously known and novel models, Ray et al. stated that, in mice, *Crb1-B* is essential and predominant, as retinal anomalies can develop even when *Crb1-A* is strictly normal. However, our human data differs from that of mouse models. In humans, the retinal degeneration appears early, and the phenotype is mainly driven by the Müller cell isoform A impairment. For instance, patients with 100% WT isoform B can develop a generalized retinal degeneration and those with the c.498_506del variant always develop a macular dystrophy. A human phenotype equivalent to the *Crb1^delB/null^* or *Crb1^delB/delB^* mouse models has never been reported, as available NGS panels do not cover the specific promotor and 5′and 3′exons of isoform B. 

### 3.4. Future Directions

In the future, human iPSC-derived retinal organoids will be required to understand the different impacts of bi-allelic *CRB1* variants. This will allow a better comprehension on the specific and combined role of *CRB1* isoforms in developmental and post adult retinae, as well as their interactions within the CRB complex. We should also screen for cases heterozygous for mutated isoform A and heterozygous or compound heterozygous for mutated isoform B. To this end, genetic analyses should be completed in patients with only one variant in *CRB1* via a novel design of the retinal NGS gene panels to accommodate screening of the specific *CRB1-B* transcript.

## 4. Materials and Methods

This study conforms to the Declaration of Helsinki and to approved protocols of the Montpellier, Lille and Rennes University hospitals. Signed informed consent for clinical examination and genetic analysis was obtained from all participants and from the parents of minors. In this retrospective study, the clinical databases of the three different national departments dedicated to inherited retinal dystrophies (Lille, Rennes, Montpellier, France) were screened to identify all patients with bi-allelic pathogenic variants in *CRB1*. The Ministry of Public Health accorded approval for biomedical research under the authorization number 11018S. 

### 4.1. Clinical Investigation

Age of onset and initial symptoms were recorded for all patients. Best-corrected visual acuity was in LogMar. Color photographs were performed with a Nidek non-mydriatic automated fundus camera (AFC 330, Nidek Inc, Tokyo, Japan). Near infrared reflectance (NIR), autofluorescence and spectral domain optical coherence tomography (SD-OCT) imaging were completed with a Combined Heidelberg Retina Angiograph + Spectralis OCT device (Heidelberg Engineering, Dossenheim, Germany). Full-field electroretinography (ff-ERG) was performed with a contact lens electrode using Ganzfeld apparatus (Ophthalmologic Monitor Metrovision, Pérenchies, France) according to the guidelines of the International Society for Electrophysiology of Vision. Goldmann visual fields were classified based on the criteria of the World Health Organization (WHO: mild visual impairment for a central visual field between 20° to 70°, low vision if limited to the central 20° and blindness if inferior to 10°).

Based on the clinical, imaging and ff-ERG data, patients were divided into one of the three following groups. The first group comprised all patients with EORD or LCA, both characterized by early onset (before the age of 5 years). In LCA, a nystagmus and oculo-digital reflexes were frequently noted within the first twelve months of life. The second group comprised patients with RCD and onset after the age of five years. The third group comprised patients with a macular dystrophy and anomalies restricted to the posterior pole and eventually the peripapillary retina (no night blindness, no constriction of peripheral visual field). 

### 4.2. SD-OCT Measurements

On SD-OCT, central foveal thickness was measured independently in each eye with automated segmentation using Heidelberg HRA software. Retinal thickness, defined as the distance between the basal retinal pigment epithelium/Bruch’s membrane and the internal limiting membrane, were measured at the fovea, at 1500 μm and 3000 μm nasal of the fovea and at 1500 μm and 3000 μm temporal of the fovea on the horizontal 9 mm long section, using the automated software and, if necessary, manually by two independent operators (KM, VS). The horizontal ellipsoid line width was measured on the horizontal 9 mm long section by two independent operators (KM, VS). Disease progression was assessed in 27 patients having undergone at least two examinations with SD-OCT. This device has only been available since 2010 in Montpellier, Lille and Nantes departments.

### 4.3. Molecular Investigation

Genomic DNA was extracted from leukocytes using a FlexiGene kit (Qiagen). We used dedicated large IRD gene panels including 230 genes associated with generalized retinal and macular dystrophies to screen for pathogenic variants [44]. The design of this panel does not allow the analysis of the specific 5′ and 3′ exons of *CRB1* isoform B.

### 4.4. Variant Analysis and Protein Modelling

Familial variant segregation analysis was carried out for each case (at least one relative). Variant classification was assessed according to the American College of Medical Genetics and Genomics guidelines [45] and using Varsome (https://varsome.com/ (accessed on 25 October 2021 )) and MobiDetails (https://mobidetails.iurc.montp.inserm.fr/MD/ (accessed on 25 October 2021)). The protein sequence of the various mutants was submitted to the webserver @TOME for sequence-structure alignment and model building using SCWRL 3.0 [46]. The Ab initio model of *CRB1* available at https://alphafold.ebi.ac.uk/entry/P82279 (accessed on 25 October 2021) was also scrutinized [47]. Resulting models were visualized and analyzed using Pymol (www.pymol.org (accessed on 25 October 2021)).

## Figures and Tables

**Figure 1 ijms-22-12642-f001:**
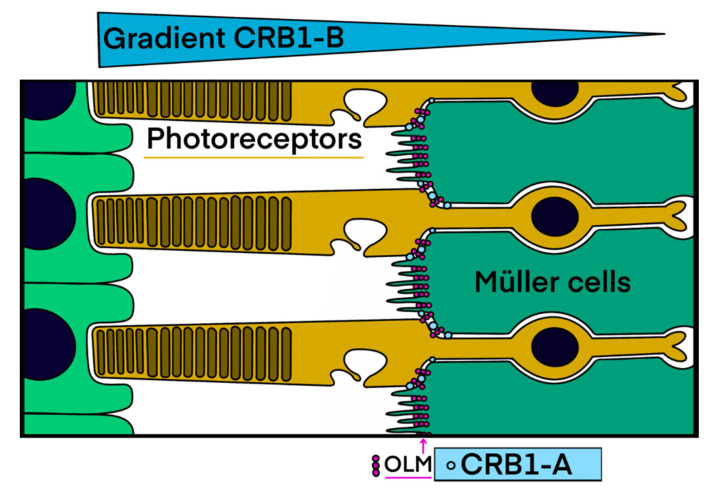
*CRB1* transmembrane isoforms: cell specificity and localization. Isoform A is located at the apical zone of Müller cells and isoform B is localized in photoreceptors with a higher expression reported in the outer segments on the side of the retinal pigment epithelium. Blue circles = *CRB1-A* transmembrane isoforms of Müller cells. Pink circles = OLM = outer limiting membrane (drawn by M.P.) formed by the adherens junctions between neighboring Müller cells and photoreceptors.

**Figure 2 ijms-22-12642-f002:**
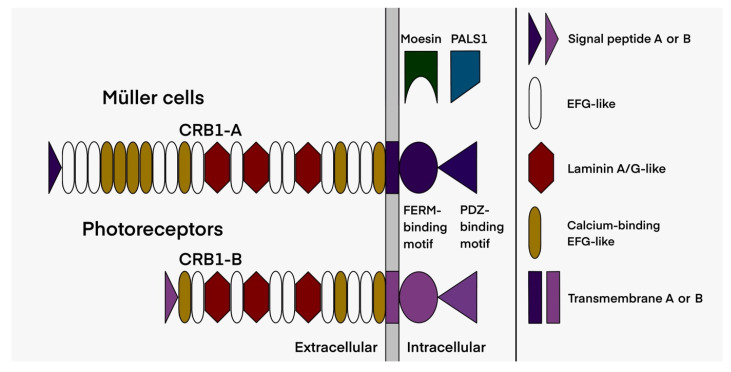
Schematic representation of the two main retinal isoforms encoded by the *CRB1* gene. Isoform B of the photoreceptors differs from the canonical isoform A of the Müller cells by a lower number of EGF-like domains, its specific signal peptide, as well as its transmembrane and intracellular domains (drawn by M.P.).

**Figure 3 ijms-22-12642-f003:**
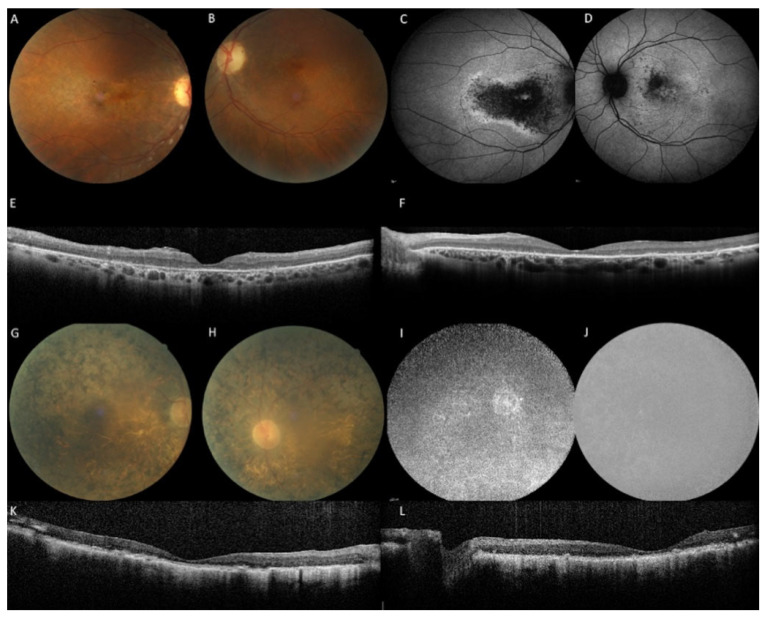
Family M-1640: the sister with an aR macular phenotype and the brother with an aR EORD. (**A**–**F**): The sister at the age of 45 years with a disease onset at 42 years complained of slow bilateral visual loss (RE 20/60, LE 20/30). No photophobia, no night blindness. Note the multiple yellow granular spots on the color frames (**A**,**B**). Peripheral retina and vessels are preserved. On fundus autofluorescence frames (FAF), the macular hypoautofluorescence is asymmetric and more extensive in the right eye. On SD-OCT scans (**E**,**F**), the macular is thickened with cystic degeneration and hyperreflective dots, in line with *CRB1*-related dystrophy. EZ and IZ lines are not visible except within the foveola in the left eye. (**G**–**L**): The brother: the visual acuity is hand motion in both eyes at the age of 46 years, night blindness appeared at 3 years, reading loss at 20 years, major difficulties to move alone at 30 years. On color frames, note the optic disc pallor, vascular attenuation and pigmentary changes of the posterior pole and the entire peripheral retina with atrophic zone. FAF (**I**,**J**) revealed severe diffuse hypoautofluorescence patches. SD-OCT (**K**,**L**) shows major alterations of the retinal segmentation with a retinal thickening and a complete loss of EZ and IZ lines.

**Figure 4 ijms-22-12642-f004:**
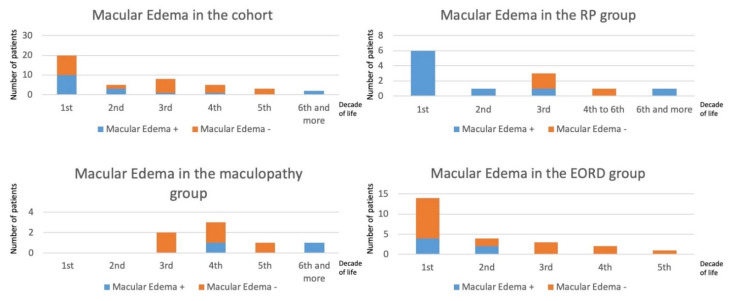
Prevalence of macular edema in EORD, RP and maculopathy by decades. The number of patients per decade is indicated in the vertical axes. The age of patients (in years) is indicated in the horizontal axes. The more severe the phenotype, the earlier the cystic changes occurred.

**Figure 5 ijms-22-12642-f005:**
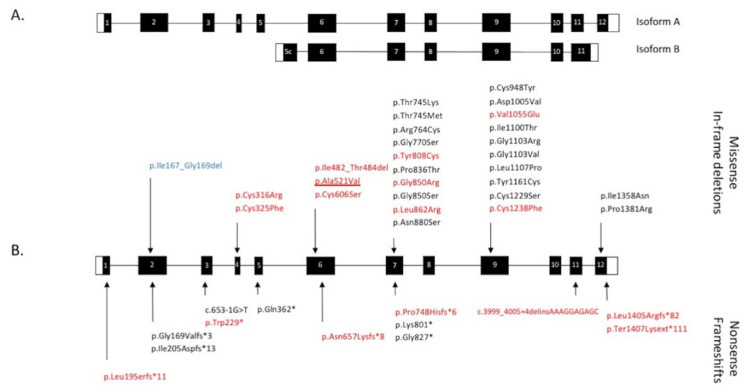
Schematic representation of the *CRB1* gene and pathogenic variants. (**A**) The 12 exons of the canonical tran-script encoding isoform A and the equivalent exons comprising the smaller transcript encoding isoform B. (**B**) Distribu-tion of *CRB1* variants (protein or cDNA level when the consequences on the protein are unknown) found in our 50-patient cohort. Missense variations and in-frame deletions are indicated above the gene structure, nonsense varia-tions and frameshifts are noted below. In red: novel variants. Underlined: specific variant responsible for both EORD and maculopathy. In blue: in-frame deletion encountered in macular dystrophy. White boxes: 5’ and 3’ UTR. Black box-es: coding parts of exons. White numbers in black boxes indicate exon number related to the canonical transcript (*CRB1-A*).

**Figure 6 ijms-22-12642-f006:**
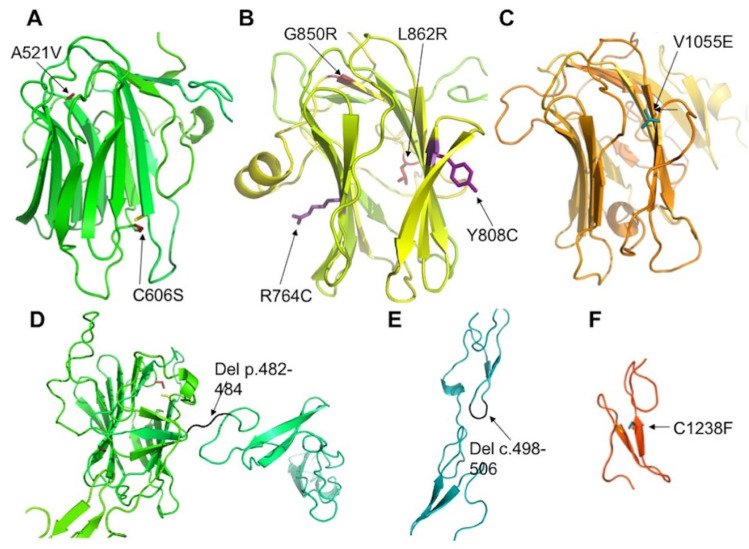
Positions of mutated residues in *CRB1*. Models of the 1st (**A**), 2nd (**B**) and 3rd (**C**) laminin domains of *CRB1* shown as colored ribbons with mutated residues indicated as sticks colored firebrick (**A**), violet (**B**) or cyan (**C**). (**D**,**E**): Deletions in EGF-like modules are shown in black. Variant p.(Cys1238Phe) (C1238F) is shown in cyan. In addition, cysteines involved in disulfide bridge with mutated cysteines are shown as yellow sticks in (**A**,**F**).

**Figure 7 ijms-22-12642-f007:**
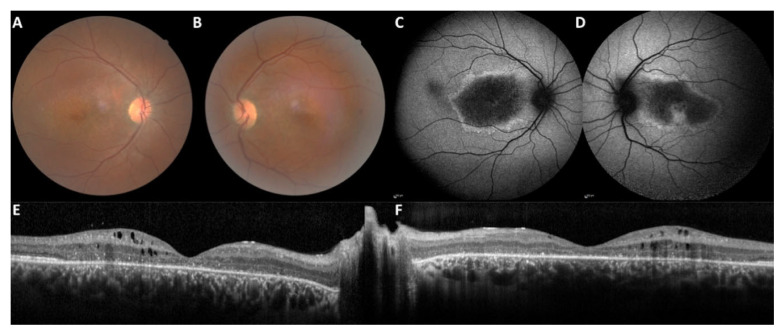
Autosomal recessive macular dystrophy in patient M-3989 carrying the c.498_506del on one allele. This patient, seen at the age of 33 years (disease onset at 17 years), had a bilateral visual loss still in progress. No photophobia, no night blindness. VA 20/400 right eye, 20/25 left eye. Note on the color frames (**A**,**B**), the multiple yellow granular macular pattern and, on the SD-OCT scan (**E**,**F**), macular thickening, cystic degeneration and hyperreflective dots in line with *CRB1*-related diseases. On FAF (**C**,**D**), the macular lesion is deeply hypoautofluorescent and surrounded by a hyperautofluorescent border. Note the foveal preservation in the left eye.

**Figure 8 ijms-22-12642-f008:**
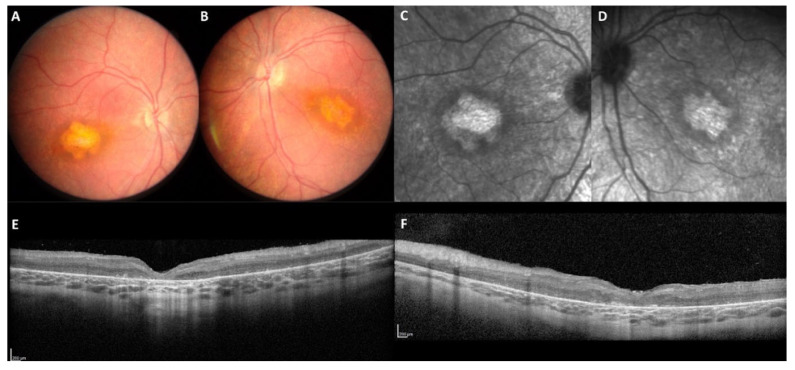
Autosomal recessive EORD in patient M-4621 homozygous for the c.4142C>G p.(Pro1381Arg) in exon 12. This patient was seen at the age of 4.5 years for low vision in both eyes. Isoform B was not impacted by the homozygous variant (WT isoform B in photoreceptors). On color frames (**A**,**B**), the macula was altered with an abnormal yellowish appearance. On infrared reflectance imaging (**C**,**D**), note the atrophic pattern of the abnormal whitish macular lesion. On the SD-OCT scans (**E**,**F**), the retina is completely thickened and disorganized, no correct segmentation is visible.

**Table 1 ijms-22-12642-t001:** Clinical and genetic data of the 50 patients with bi-allelic pathogenic variants in *CRB1* classified according to phenotype. Early onset retinal dystrophy (EORD) in 28 patients, retinitis pigmentosa (RP; or rod-cone dystrophy) in 13 patients and macular dystrophy (MD) in 9 patients. HM: hand motion, LP: light perception; ND: not done; PPARPE: preserved para-arteriolar retinal pigment epithelium.

Patient Number (Gender)	Variant 2 cDNA Protein	Age at Onset Age at First Visit	Refraction (SE) OD OG	Variant 1 cDNA Protein	Initial BCVA (logMAR, Snellen) OD OG	Final BCVA (logMAR, Snellen) OD OG	Follow up (Months) Age at Final Visit	Phenotype	Optic Disc Drusen	PPARPE	Coats-Like	Macular Oedema
M-1640-1	c.1562C>T	3	2.25	c.1562C>T	2.3 (HM)	2.3 (HM)	82	EORD	No	No	No	No
(M)	p.Ala521Val	46.5	2	p.Ala521Val	2.3 (HM)	2.6 (LP)	53					
M-2129	c.2234C>T	1.5	3.25	c.2234C>T	0.3 (20/40)	0.8 (20/125)	66	EORD	No	Yes	No	No
(M)	p.Thr745Met	9.5	3.25	p.Thr745Met	1.0 (20/200)	0.8 (20/125)	15					
M-567	c.1445_1453del	3	−0.25	c.2290C>T	0 (20/20)	0.7 (20/100)	84	EORD	Yes	No	No	Yes
(M)	p.Ile482_Thr484del	4	−0.50	p.Arg764Cys	0 (20/20)	1.3 (20/400)	11					
M-116	c.4219T>A	5	2	c.687G>A	0.7 (20/100)	ND	ND	EORD	ND	ND	ND	ND
(F)	p.Ter1407Lysext*111	43.5	2	p.Trp229*	0.7 (20/100)							
M-3361	c.2290C>T	4.5	2.5	c.2290C>T	0.1 (20/25)	0.3 (20/40)	25	EORD	Yes	No	No	Yes
(M)	p.Arg764Cys	5	3.5	p.Arg764Cys	0.4 (20/50)	0.3 (20/40)	7.5					
M-3591	c.2401A>T	1	2.75	c.2290C>T	0.7 (20/100)	0.3 (20/40)	23	EORD	No	Yes	No	Yes
(M)	p.Lys801*	4.5	3.25	p.Arg764Cys	0.7 (20/100)	0.3 (20/40)	6					
M-2897	c.2843G>A	5	6.25	c.2548G>A	2.3 (HM)	ND	ND	EORD	No	No	No	No
(F)	p.Cys948Tyr	34	6	p.Gly850Ser	1.7 (20/1000)							
M-3235	c.4214del	4	0.5	c.2290C>T	0.7 (20/100)	ND	ND	EORD	Yes	Yes	No	Yes
(M)	p.Leu1405Argfs*82	17.5	1.75	p.Arg764Cys	0.7 (20/100)							
M-1316-4	c.2243del	5	3.75	c.2243del	0.4 (20/50)	ND	ND	EORD	No	Yes	No	No
(F)	p.Pro748Hisfs*6	7	1.75	p.Pro748Hisfs*6	0.7 (20/100)							
M-340	c.2479G>T	1	3.75	c.2290C>T	1.0 (20/200)	2.3 (HM)	200	EORD	Yes	Yes	No	No
(F)	p.Gly827*	5	3.25	p.Arg764Cys	1.0 (20/200)	1.3 (20/400)	22					
L- 15090708 (M)	c.3713G>T	1	10	c.2401A>T	1.3 (20/400)	1.1 (20/250)	48	EORD	No	Yes	No	No
	p.Cys1238Phe	4	10	p.Lys801*	1.0 (20/200)	1.1 (20/250)	8					
L-14061710	c.4142C>G	1	7	c.2548G>A	0.7 (20/100)	0.4 (20/50)	84	EORD	Yes	No	No	No
(F)	p.Pro1381Arg	4	7	p.Gly850Ser	0.7 (20/100)	0.4 (20/50)	11					
XZ-372875	c.3320T>C	4	3	c.3686G>C	0.8 (20/125)	0.7 (20/100)	72	EORD	No	Yes	No	No
(F)	p.Leu1107Pro	4	4	p.Cys1229Ser	0.9 (20/160)	0.7 (20/100)	10					
M-1316-1	c.2243del	3.5	3.75	c.2243del	1.2 (20/320)	2.3 (HM)	116	EORD	No	Yes	No	No
(F)	p.Pro748Hisfs*67	4.5	1.75	p.Pro748Hisfs*6	1.0 (20/200)	1.3 (20/400)	17					
M-3324	c.2401A>T	1.5	ND	c.1084C>T	2.3 (HM)	ND	ND	EORD	No	No	No	No
(F)	p.Lys801*	27	ND	p.Gln362*	2.3 (HM)							
M-2427	c.4073T>A	<5	7.5	c.4073T>A	2.3 (HM)	2.0 (20/2000)	26	EORD	No	No	No	No
(M)	p.Ile1358Asn	5	8	p.Ile1358Asn	2.3 (HM)	2.0 (20/2000)	7					
M-4621-1	c.4142C>G	2.5	5.75	c.4142C>G	ND	0.8 (20/125)	21	EORD	No	Yes	No	No
(F)	p.Pro1381Arg	3	6.5	p.Pro1381Arg	ND	0.8 (20/125)	4.5					
M-1580	c.3320T>C	1	4.5	c.3164T>A	2.3 (HM)	2.6 (LP)	47	EORD + nystagmus	Yes	Yes	No	No
(F)	p.Leu1107Pro	17	5.25	p.Val1055Glu	2.3 (HM)	2.6 (LP)	21					
M-2123	c.3999_4005+4delinsAAAGGAGAGC	0.5	ND	c.3999_4005+4delinsAAAGGAGAGC	ND	ND	ND	EORD + nystagmus	No	No	No	No
(M)	p.?	0.5	ND	p.?	ND	ND						
M-2123-2	c.3999_4005+4delinsAAAGGAGAGC	0.4	4	c.3999_4005+4delinsAAAGGAGAGC	ND	ND	ND	EORD + nystagmus	No	No	No	No
(M)	p.?	0.5	4.5	p.?	ND	ND						
M-2186	c.1971del	0.5	3.5	c.1971del	2.3 (HM)	2.3 (HM)	76	EORD + nystagmus	No	No	No	ND
(F)	p.Asn657Lysfs*8	1	5	p.Asn657Lysfs*8	2.3 (HM)	2.3 (HM)	7					
M-2184	c.2843G>A	4	3.75	c.2401A>T	1.0 (20/200)	1.3 (20/400)	82	EORD + nystagmus	Yes	Yes	Yes	Yes
(F)	p.Cys948Tyr	9	4.25	p.Lys801*	1.3 (20/400)	1.3 (20/400)	16					
XZ-301265-1 (F)	c.2291G>A	5	7.75	c.2291G>A	2.3 (HM)	2.3 (HM)	60	EORD + nystagmus	No	No	No	No
	p.Arg764His	26	5.75	p.Arg764His	2.3 (HM)	2.3 (HM))	31					
XZ-301265-2 (M)	c.2291G>A	ND	6	c.2291G>A	2.6 (LP)	2.6 (LP)	96	EORD + nystagmus	No	No	No	No
	p.Arg764His	34	6	p.Arg764His	2.6 (LP)	2.6 (LP)	42					
XZ-010621	c.2234C>A	0.5	8.75	c.2843G>A	0.7 (20/100)	0.9 (20/160)	228	EORD + nystagmus	No	No	No	Yes
(M)	p.Thr745Lys	12	8.75	p.Cys948Tyr	0.5 (20/63)	0.6 (20/80)	31					
XZ-344444	c.2498G>A	0.5	9.5	c.506del	2.6 (LP)	2.6 (LP)	24	EORD + nystagmus	Yes	No	No	No
(F)	p.Gly833Asp	28	8.5	p.Gly169Valfs*3	2.6 (LP)	2.6 (LP)	30					
M-1861	c.2639A>G	5	0.25	c.54_55insT	1.5 (20/600)	ND	ND	EORD	Yes	Yes	No	No
(M)	p.Asn880Ser	17	0	p.Leu19Serfs*11	1.2 (20/320)							
M-1732	c.2401A>T	3	9.25	c.974G>T	0.7 (20/100)	0.7 (20/100)	39	EORD	Yes	No	No	No
(F)	p.Lys801*	23.5	8.5	p.Cys325Phe	0.7 (20/100)	0.7 (20/100)	26.5					
M-3530	c.2234C>T	3	2	c.2234C>T	1.0 (20/200)	1.0 (20/200)	36	RP	Yes	Yes	No	No
(M)	p.Thr745Met	25	2.5	p.Thr745Met	1.3 (20/400)	1.0 (20/200)	28					
XZ-381491	c.2843G>A	9	7.75	c.3307G>A	0.6 (20/80)	2.6 (LP)	12	RP	No	No	No	No
(F)	p.Cys948Tyr	26	7	p.Gly1103Arg	0.9 (20/160)	2.6 (LP)	27					
M-731	c.2290C>T	6	4	c.3308G>T	0.5 (20/63)	0.8 (20/125)	182	RP	No	Yes	No	Yes
(M)	p.Arg764Cys	8	4.25	p.Gly1103Val	0.4 (20/50)	0.7 (20/100)	23					
M-2804-1	c.2308G>A	9	1.5	c.2308G>A	0.4 (20/50)	0.4 (20/50)	32	RP	No	No	No	Yes
(M)	p.Gly770Ser	9.5	1	p.Gly770Ser	0.3 (20/40)	0.4 (20/50)	12					
M-2804-2	c.2308G>A	7	ND	c.2308G>A	ND	ND	ND	RP	No	No	No	Yes
(F)	p.Gly770Ser	32		p.Gly770Ser								
M-2121	c.1817G>C	14	5.5	c.613_619del	0.4 (20/50)	0.4 (20/50)	37	RP	No	No	No	Yes
(M)	p.Cys606Ser	17.5	5.5	p.Ile205Aspfs*13	0.4 (20/50)	0.4 (20/50)	21					
M-2415	c.2843G>A	6	−0.25	c.2423A>G	0.2 (20/32)	0.7 (20/100)	54	RP	Yes	No	No	Yes
(F)	p.Cys948Tyr	23.5	−0.25	p.Tyr808Cys	0.2 (20/32)	0.4 (20/50)	28					
M-699	c.2290C>T	7	ND	c.2290C>T	2.6 (LP)	ND	ND	RP	ND	ND	ND	ND
(F)	p.Arg764Cys	80		p.Arg764Cys	2.6 (LP)							
M-4075	c.3299>C	6	0.75	c.653-1G>T	0.4 (20/50)	ND	ND	RP	No	No	No	Yes
(F)	p.Ile1100Thr	7	0.75	p.?	0.1 (20/25)							
M-4463	c.2843G> A	6	ND	c.2843G> A	2.6 (LP)	ND	ND	RP	No	No	No	No
(F)	p. Cys948Tyr	48	ND	p.Cys948Tyr	2.6 (LP)							
M-4570-1	c.2506C>A	6	3	c.2290C>T	0.2 (20/32)	0.2 (20/32)	29	RP	No	No	No	Yes
(F)	p.Pro836Thr	10	3	p.Arg764Cys	0.5 (20/63)	0.7 (20/100)	12					
M–3915	c.613_619del	>5	ND	c.3482A>G	0.7 (20/100)	ND	ND	RP	No	No	No	Yes
(M)	p.Ile205Aspfs*13	61.5	−0.25	p.Tyr1161Cys	0.2 (20/32)							
XZ-332989	c.2549G>C	5	5	c.2585T>G p.Leu862Arg	0.6 (20/80)	0.8 (20/125)	108	RP	No	No	No	Yes
(F)	p.Gly850Arg	7	5.25		0.6 (20/80)	0.8 (20/125)	16					
L-93112991 (M)	c.2843G>A	ND	−1.75	c.498_506del	0.5 (20/63)	ND	ND	MD	No	No	No	ND
	p.Cys948Tyr	44	−1.50	p.Ile167_Gly169del	0.7 (20/100)							
L-13010924 (M)	c.2843G>A	22	0.75	c.498_506del	0.6 (20/80)	1.5 (20/600)	60	MD	No	No	No	No
	p.Cys948Tyr	23	0.25	p.Ile167_Gly169del	0.2 (20/32)	0.4 (20/50)	28					
M-1640-4	c.1562C>T	42	−1.00	c.1562C>T	0.7 (20/100)	0.5 (20/63)	36	MD	No	No	No	No
(F)	p.Ala521Val	42	−1.25	p.Ala521Val	1.0 (20/200)	0.2 (20/32)	45					
M-3989	c.2843G>A	17	−0.50	c.498_506del	1.3 (20/400)	ND	ND	MD	No	No	No	Yes
(M)	p.Cys948Tyr	33	−1.50	p.Ile167_Gly169del	0.1 (20/25)							
M-2183	c.946T>C	21	3.25	c.498_506del	2.3 (HM)	ND	ND	MD	No	No	No	No
(M)	p.Cys316Arg	33	3.25	p.Ile167_Gly169del	2.3 (HM)							
M-3073-1	c.2401A>T	34	−1.00	c.498_506del	0.5 (20/63)	ND	ND	MD	No	No	No	No
(M)	p.Lys801*	37	−0.50	p.Ile167_Gly169del	0.4 (20/50)							
M-3073-3	c.2401A>T	ND	ND	c.498_506del	ND	ND	ND	MD	ND	ND	ND	ND
(F)	p.Lys801*	46		p.Ile167_Gly169del								
M-3377	c.3014A>T	40	−0.50	c.498_506del	1.3 (20/400)	ND	ND	MD	No	No	No	Yes
(M)	p.Asp1005Val	53.5	−0.75	p.Ile167_Gly169del	1.3 (20/400)							
M-3343	c.498_506del	16	0.5	c.4142C>G	0.4 (20/50)	ND	ND	MD	No	No	No	No
(M)	p.Ile167_Gly169del	29.5	0.5	p.Pro1381Arg	0.7 (20/100)							

**Table 2 ijms-22-12642-t002:** SD-OCT macular measurements. Retinal thickness in microns. EORD: early onset retinal dystrophy. Controls were matched by age and sex. SD = standard deviation. T1500 and T3000 = retinal thickness at 1500 and 3000 μ temporal of the fovea, N1500 = retinal thickness at 1500 μ nasal of the fovea. * SD not calculated because measurement was made only on one eye.

		N Eyes	Fovea Mean ± (SD)	T1500 Mean ± (SD)	T3000 Mean ± (SD)	N1500 Mean ± (SD)
**Controls**	**Global**	**28**	**226.79 (±20.71)**	**321.46 (±19.03)**	**257.89 (±17.08)**	**347.21 (±19.29)**
	**[0;10] years**	**0**	**-**	**-**	**-**	**-**
	**]10;20] years**	**2**	**266.50 (±0.71)**	**310.00 (±2.83)**	**260.00 (±1.41)**	**341.00 (±1.41)**
	**]20;30] years**	**10**	**218.30 (±14.58)**	**322.90 (±18.04)**	**261.40 (±12.31)**	**352.50 (±21.08)**
	**]30;40] years**	**4**	**234.25 (±20.65)**	**305.75 (±6.55)**	**242.50 (±2.38)**	**330.25 (±13.74)**
	**]40;50] years**	**4**	**204.25 (±7.32)**	**312.25 (±6.70)**	**244.25 (±7.37)**	**337.50 (±9.54)**
	**>50 years**	**8**	**235.00 (±14.92)**	**335.00 (±22.37)**	**267.50 (±23.19)**	**355.50 (±19.89)**
**Maculopathy**	**Global**	**14**	**106.57 (±23.64)**	**284.43 (±47.00)**	**232.36 (±28.32)**	**308.79 (±33.24)**
	**[0;10] years**	**0**	**-**	**-**	**-**	**-**
	**]10;20] years**	**0**	**-**	**-**	**-**	**-**
	**]20;30] years**	**4**	**108.00 (±5.29)**	**248.50 (±31.27)**	**214.00 (±28.01)**	**281.00 (±14.88)**
	**]30;40] years**	**6**	**96.33 (±21.92)**	**299.67 (±45.45)**	**255.00 (±20.28)**	**335.00 (±23.16)**
	**]40;50] years**	**2**	**108.00 (±1.41)**	**263.00 (±9.90)**	**208.50 (±2.12)**	**304.00 (±36.77)**
	**>50 years**	**2**	**133.00 (±52.33)**	**332.00 (±57.98)**	**225.00 (±18.38)**	**290.50 (±41.72)**
**Retinitis** **Pigmentosa**	**Global**	**21**	**272.95 (±150.51)**	**353.48 (±49.72)**	**344.76 (±75.15)**	**382.67 (±42.48)**
	**[0;10] years**	**8**	**296.50 (±211.71)**	**360.88 (±67.27)**	**358.38 (±100.29)**	**396.88 (±53.90)**
	**]10;20] years**	**4**	**266.75 (±112.23)**	**356.75 (±12.34)**	**343.00 (±38.76)**	**372.00 (±21.02)**
	**]20;30] years**	**6**	**242.67 (±115.77)**	**362.50 (±35.14)**	**364.83 (±58.33)**	**384.17 (±38.92)**
	**]30;40] years**	**0**	**-**	**-**	**-**	**-**
	**]40;50] years**	**1**	**177.00 ***	**257.00 ***	**250.00 ***	**314.00 ***
	**>50 years**	**2**	**330.00 (±93.34)**	**338.50 (±20.51)**	**281.00 (±4.24)**	**377.00 (±0.00)**
**EORD**	**Global**	**17**	**268.94 (±169.17)**	**363.71 (±55.83)**	**332.53 (±58.25)**	**415.29 (±70.12)**
	**[0;10] years**	**10**	**290.30 (±201.05)**	**388.10 (±51.25)**	**355.10 (±61.94)**	**434.00 (±71.05)**
	**]10;20] years**	**2**	**403.00 (±21.21)**	**342.50 (±3.54)**	**318.00 (±12.73)**	**464.50 (±6.36)**
	**]20;30] years**	**2**	**190.50 (±61.52)**	**374.00 (±18.38)**	**324.50 (±30.41)**	**404.50 (±27.58)**
	**]30;40] years**	**1**	**185.00 ***	**311.00 ***	**257.00 ***	**348.00 ***
	**]40;50] years**	**0**	**-**	**-**	**-**	**-**
	**>50 years**	**2**	**148.50 (±31.82)**	**279.00 (±41.01)**	**280.00 (±42.43)**	**317.00 (±1.41)**

**Table 3 ijms-22-12642-t003:** Pathogenic variants in *CRB1* in each affected patient and probable impact on specific Müller cell and photoreceptor isoforms. From left to the right: Patient ID; allele 1 (protein annotation whenever possible), impact on isoform A and isoform B (protein domain level) from allele 1; allele 2, impact on isoform A and isoform B from allele 2 (protein domain level); final impact on protein *CRB1* in Müller cells and final impact on *CRB1* in photoreceptors (both in terms of amount of WT and mutated proteins). WT = wild type. NMD = presumed nonsense-mediated decay when stop codon or frameshift is present before the penultimate exon.

Patient ID	Allele 1	Impact on *CRB1-A*	Impact on *CRB1-B*	Allele 2	Impact on *CRB1-A*	Impact on *CRB1-B*	Müller Cell Isoform	Photoreceptor Isoform
M-1640-1	p.Ala521Val	Laminin G-like 1	Laminin G-like 1	p.Ala521Val	Laminin G-like 1	Laminin G-like 1	100% mutated	100% mutated
M-2129	p.Thr745Met	Laminin G-like 2	Laminin G-like 2	p.Thr745Met	Laminin G-like 2	Laminin G-like 2	100% mutated	100% mutated
M-567	p.Arg764Cys	Laminin G-like 2	Laminin G-like 2	p.Ile482_Thr484del	Deletion between EGF-like 11 and Lam G-like 1	Deletion between EGF-like 11 and Lam G-like 1	100% mutated	100% mutated
M-116	p.Trp229*	NMD	WT	p.Ter1407Lysext*111	Cytoplasmic C-term	WT	50% mutated/0% WT	100% WT
M-3361	p.Arg764Cys	Laminin G-like 2	Laminin G-like 2	p.Arg764Cys	Laminin G-like 2	Laminin G-like 2	100% mutated	100% mutated
M-3591	p.Arg764Cys	Laminin G-like 2	Laminin G-like 2	p.Lys801*	NMD	NMD	50% mutated/0% WT	50% mutated/0% WT
M-2897	p.Gly850Ser	Laminin G-like 2	Laminin G-like 2	p.Cys948Tyr	EGF-like 14	EGF-like 14	100% mutated	100% mutated
M-3235	p.Arg764Cys	Laminin G-like 2	Laminin G-like 2	p.Leu1405Argfs*82	Cytoplasmic C-term	WT	100% mutated	50% mutated/50% WT
M-1316-4	p.Pro748Hisfs*6	NMD	NMD	p.Pro748Hisfs*6	NMD	NMD	No protein	No protein
M-340	p.Arg764Cys	Laminin G-like 2	Laminin G-like 2	p.Gly827*	NMD	NMD	50% mutated/0% WT	50% mutated/0% WT
L- 15090708	p.Lys801*	NMD	NMD	p.Cys1238Phe	EGF-like-17	EGF-like-17	50% mutated/0% WT	50% mutated/0% WT
L-14061710	p.Gly850Ser	Laminin G-like 2	Laminin G-like 2	p.Pro1381Arg	Cytoplasmic C-term	WT	100% mutated	50% mutated/50% WT
XZ-372875	p.Cys1229Ser	EGF-like 19	EGF-like 19	p.Leu1107Pro	Laminin G-like 3	Laminin G-like 3	100% mutated	100% mutated
M-1316-1	p.Pro748Hisfs*6	NMD	NMD	p.Pro748Hisfs*6	NMD	NMD	No protein	No protein
M-3324	p.Gln362*	NMD	WT	p.Lys801*	NMD	NMD	No protein	0% mutated/50% WT
M-2427	p.Thr745Met	Laminin G-like 2	Laminin G-like 2	p.Ile1358Asn	Cytoplasmic C-term	WT	100% mutated	50% mutated/50% WT
M-4621-1	p.Pro1381Arg	Cytoplasmic C-term	WT	p.Pro1381Arg	Cytoplasmic C-term	WT	100% mutated	100% WT
M-1580	p.Val1055Glu	Laminin G-like 3	Laminin G-like 3	p.Leu1107Pro	Laminin G-like 3	Laminin G-like 3	100% mutated	100% mutated
M-2123	c.3999_4005 + 4delinsAAAGGAGAGC	?	Cytoplasmic C-term	c.3999_4005 + 4delinsAAAGGAGAGC	?	Cytoplasmic C-term	100% mutated	100% mutated
M-2123-2	c.3999_4005 + 4delinsAAAGGAGAGC	?	Cytoplasmic C-term	c.3999_4005 + 4delinsAAAGGAGAGC	?	Cytoplasmic C-term	100% mutated	100% mutated
M-2186	p.Asn657Lysfs*8	NMD	NMD	p.Asn657Lysfs*8	NMD	NMD	No protein	no protein
M-2184	p.Lys801*	NMD	NMD	p.Cys948Tyr	EGF-like 14	EGF-like 14	50% mutated/0% WT	50% mutated/0% WT
XZ-301265-1	p.Arg764Cys	Laminin G-like 2	Laminin G-like 2	p.Arg764Cys	Laminin G-like 2	Laminin G-like 2	100% mutated	100% mutated
XZ-301265-2	p.Arg764Cys	Laminin G-like 2	Laminin G-like 2	p.Arg764Cys	Laminin G-like 2	Laminin G-like 2	100% mutated	100% mutated
XZ-010621	p.Cys948Tyr	EGF-like 14	EGF-like 14	p.Thr745Lys	Laminin G-like 2	Laminin G-like 2	100% mutated	100% mutated
XZ-344444	p.Gly169Valfs*3	NMD	WT	p.Gly833Asp	Laminin G-like 2	Laminin G-like 2	50% mutated/0% WT	50% mutated/50% WT
M-1861	p.Leu19Serfs*11	NMD	WT	p.Asn880Ser	Laminin G-like 2	Laminin G-like 2	50% mutated/0% WT	50% mutated/50% WT
M-1732	p.Cys325Phe	EGF-like 8	WT	p.Lys801*	NMD	NMD	50% mutated/0% WT	0% mutated/50% WT
M-3530	p.Thr745Met	Laminin G-like 2	Laminin G-like 2	p.Thr745Met	Laminin G-like 2	Laminin G-like 2	100% mutated	100% mutated
XZ-381491	p.Gly1103Arg	Laminin G-like 3	Laminin G-like 3	p.Cys948Tyr	EGF-like 14	EGF-like 14	100% mutated	100% mutated
M-731	p.Gly1103Val	Laminin G-like 3	Laminin G-like 3	p.Arg764Cys	Laminin G-like 2	Laminin G-like 2	100% mutated	100% mutated
M-2804-1	p.Gly770Ser	Laminin G-like 2	Laminin G-like 2	p.Gly770Ser	Laminin G-like 2	Laminin G-like 2	100% mutated	100% mutated
M-2804-2	p.Gly770Ser	Laminin G-like 2	Laminin G-like 2	p.Gly770Ser	Laminin G-like 2	Laminin G-like 2	100% mutated	100% mutated
M-2121	p.Ile205Aspfs*13	NMD	WT	p.Cys606Ser	Laminin G-like 1	Laminin G-like 1	50% mutated/0% WT	50% mutated/50% WT
M-2415	p.Tyr808Cys	Laminin G-like 2	Laminin G-like 2	p.Cys948Tyr	EGF-like 14	EGF-like 14	100% mutated	100% mutated
M-699	p.Arg764Cys	Laminin G-like 2	Laminin G-like 2	p.Arg764Cys	Laminin G-like 2	Laminin G-like 2	100% mutated	100% mutated
M-4075	c.653-1G > T	?	WT	p.Ile1100Thr	Laminin G-like 3	Laminin G-like 3	Unknown	50% mutated/50% WT
M-4463	p.Cys948Tyr	EGF-like 14	EGF-like 14	p. Cys948Tyr	EGF-like 14	EGF-like 14	100% mutated	100% mutated
M-4570-1	p.Arg764Cys	Laminin G-like 2	Laminin G-like 2	p.Pro836Thr	Laminin G-like 2	Laminin G-like 2	100% mutated	100% mutated
M-3915	p.Tyr1161Cys	EGF-like 15	EGF-like 15	p.Ile205Aspfs*13	NMD	WT	50% mutated/0% WT	50% mutated/50% WT
XZ-332989	p.Leu862Arg	Laminin G-like 2	Laminin G-like 2	p.Gly850Arg	Laminin G-like 2	Laminin G-like 2	100% mutated	100% mutated
L-93112991	p.Ile167_Gly169del	EGF-like 4	WT	p.Cys948Tyr	EGF-like 14	EGF-like 14	100% mutated	50% mutated/50% WT
L-13010924	p.Ile167_Gly169del	EGF-like 4	WT	p.Cys948Tyr	EGF-like 14	EGF-like 14	100% mutated	50% mutated/50% WT
M-1640-4	p.Ala521Val	Laminin G-like 1	Laminin G-like 1	p.Ala521Val	Laminin G-like 1	Laminin G-like 1	100% mutated	100% mutated
M-3989	p.Ile167_Gly169del	EGF-like 4	WT	p.Cys948Tyr	EGF-like 14	EGF-like 14	100% mutated	50% mutated/50% WT
M-2183	p.Ile167_Gly169del	EGF-like 4	WT	p.Cys316Arg	EGF-like 8	WT	100% mutated	100% WT
M-3073-1	p.Ile167_Gly169del	EGF-like 4	WT	p.Lys801*	NMD	NMD	50% mutated/0% WT	0% mutated/50% WT
M-3073-3	p.Ile167_Gly169del	EGF-like 4	WT	p.Lys801*	NMD	NMD	50% mutated/0% WT	0% mutated/50% WT
M-3377	p.Ile167_Gly169del	EGF-like 4	WT	p.Asp1005Val	Laminin G-like 3	Laminin G-like 3	100% mutated	50% mutated/50% WT
M-3343	p.Ile167_Gly169del	EGF-like 4	WT	p.Pro1381Arg	Cytoplasmic C-term	WT	100% mutated	100% WT

**Table 4 ijms-22-12642-t004:** Novel *CRB1* pathogenic variants found in our 50-patient cohort. For each variant, we indicate the cDNA and protein position relative to the transcript NM_201253.3, the exon number, the type of variant (indel, insertion and deletion of one or several nucleotides, Del, deletion), the frequency of the variant in the database gnomAD, the result of in silico pathogenicity predictions (pathogenic versus benign predictions) and number of programs, the ACMG classification (P, pathogenic; LP, likely pathogenic) and the domain impacted in the canonical protein (TM, transmembrane domain).

cDNA	Protein	Exon	Type of Variant	Gnomad	Softwares Prediction	ACMG Classification	Domain
c.54_55insT	p.Leu19Serfs*11	1	Indel	Absent	Damaging	P	Signal peptide
c.687G>A	p.Trp229*	3	Nonsense	Absent	Damaging	P	EGF-like-6
c.946T>C	p.Cys316Arg	4	Missense	Absent	Damaging for 23/1	LP	EGF-like-8
c.974G>T	p.Cys325Phe	4	Missense	Absent	Damaging for 22/2	LP	EGF-like-8
c.1445_1453del	p.Ile482_Thr484del	6	Del	Absent	Damaging	LP	ND
c.1562C>T	p.Ala521Val	6	Missense	Absent	Damaging for 8/15	LP	Laminin G-like-1
c.1817G>C	p.Cys606Ser	6	Missense	Absent	Damaging for 8/16	LP in *trans*	Laminin G-like-1
c.1971del	p.Asn657Lysfs*8	6	Del	Absent	Damaging	P	Laminin G-like-1
c.2243del	p.Pro748Hisfs*6	7	Del	Absent	Damaging	P	Laminin G-like-2
c.2423A>G	p.Tyr808Cys	7	Missense	Absent	Damaging for 6/18	LP in *trans*	Laminin G-like-2
c.2549G>C	p.Gly850Arg	7	Missense	Absent	Damaging for 21/3	LP	Laminin G-like-2
c.2585T>G	p.Leu862Arg	7	Missense	Absent	Damaging for 22/2	LP	Laminin G-like-2
c.3164T>A	p.Val1055Glu	9	Missense	Absent	Damaging for 18/6	LP	Laminin G-like-3
c.3713G>T	p.Cys1238Phe	9	Missense	Absent	Damaging for 23/1	LP	EGF-like-17
c.3999_4005 + 4delinsAAAGGAGAGC	p.?	11	Indel	Absent	Damaging	P	EGF-like-18/TM domain
c.4214del	p.Leu1405Argfs*82	12	Del	Absent	Damaging	LP	Intracellular tail
c.4219T > A	p.Ter1407Lysext*111	12	Stop loss	Absent	Damaging for 3/5	LP in *trans*	Intracellular tail

## Data Availability

All data are contained within the article.

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
