# Peer review of "CRB1*-Related Retinal Dystrophies in a Cohort of 50 Patients: A Reappraisal in the Light of Specific Müller Cell and Photoreceptor *CRB1* Isoforms"

_ijms, 2021, doi:10.3390/ijms222312642_

Round 1

Reviewer 1 Report

The article titled: “ CRB1-related retinal dystrophies in a cohort of 50 patients: a reappraisal in the light of specific Müller cell  and photoreceptor CRB1 isoforms “ by  Kévin Mairot et al. shows how CRB1-related dystrophies in humans, which should be taken into consideration for future clinical trials. The authors have studied different Retinal Degeneration Diseases and the role of CRB1 in them.

Some comments about the work:

-Line 68: Please include the RCG failure into IRD.

-Line 121: Please define OLM

-The number of lines is disappeared

-Page 10:  the authors write:” we observed a macular thickening in line with CRB1 lamination anomalies”. Could you explain these alterations?

-Figure 4: Please, include a vertical axis title. Please in the horizontal axis, the units should be included.

-Page 20:  Please define NGS.

-Page 20: The authors have included OLM in the figure 1, but there is no reference to its role, or function or implication in the article.

On page 20, the authors re-write OLM, please define it, and if it is possible include a phrase in the first of the text.

Page 20: The authors write:” Recently, human induced pluripotent stem cells [43] were generated from a patient with a macular dystrophy carrying the c.498_506del in trans with the c.613_619del variant in the aim of studying the dysfunction and changes in Müller cells and photoreceptors in human retinal organoids”. The text is adequate, but a specific discussion phrase is needed for the article.

Author Response

Reviewer 1:

Minor revisions

  1. Line 68: Please include the RCG failure into IRD.

            Answer: The disease group IRDs are classically defined by a degeneration of the photoreceptors, either as a primary defect or secondary to RPE degeneration. RCG failure is not associated with IRDs but with Inherited Optic Neuropathy (ION). To avoid confusion, as it was beyond the focus of the article, we did not extend the introduction to include IONs.

  1. Line 121: Please define OLM

Answer. We defined OLM as follows: “OLM = Outer limiting membrane (drawn by M.P.) formed by the adherens junctions between neighboring Müller cells and photoreceptors.” (lines 121-123)

  1. The number of lines is disappeared.

Answer: This has been corrected.

  1. Page 10:  the authors write:” we observed a macular thickening in line with CRB1 lamination anomalies”. Could you explain these alterations?

            Answer: lines 177-178: The phrase has been modified to “, we observed a macular thickening with loss of the retinal reflectivity segmentation pattern.

  1. Figure 4: Please, include a vertical axis title. Please in the horizontal axis, the units should be included.

            Answer: Vertical and horizontal axes were added. The legend has been modified (lines198-200) as follows: Figure 4. Prevalence of macular edema in EORD, RP and maculopathy by decades. The number of patients per decade is indicated in the vertical axes. The age of patients in years is indicated in the horizontal axes. The more severe the phenotype, the earlier the cystic changes occurred.

  1. Page 20:  Please define NGS.

            Answer: Line 349, NGS has been defined as “next generation sequencing (NGS)”

  1. Page 20: The authors have included OLM in the figure 1, but there is no reference to its role, or function or implication in the article.

Answer: The OLM has been better defined in the figure legend, as mentioned above.

  1. On page 20, the authors re-write OLM, please define it, and if it is possible include a phrase in the first of the text.

Answer: lines 354-357. The corresponding sentence in the discussion has been modified to: “These cell-surface proteins are both present at the level of the outer limiting membrane (OLM), which is formed by the adherens junctions between the adjacent Müller cells and photoreceptors, creating a distinct barrier between the neural retina and the inner/outer segments.

  1. Page 20: The authors write:” Recently, human induced pluripotent stem cells [43] were generated from a patient with a macular dystrophy carrying the c.498_506del in trans with the c.613_619del variant in the aim of studying the dysfunction and changes in Müller cells and photoreceptors in human retinal organoids”. The text is adequate, but a specific discussion phrase is needed for the article.

Answer: Lines 377-378: We added the following sentence: “These patient-specific organoids will provide valuable insights into the macular pathophysiology specifically associated with the c.498_506del variant.”. 

As requested by MDPI, we have added the following information in the material and methods section (lines 503-505): “The Ministry of Public Health accorded approval for biomedical research under the authorization number 11018S.”

   We hope these modifications meet with your approval. We thank you in advance for your time and look forward to hearing from you.

Reviewer 2 Report

Congratulations. The work is characterized by an appropriate selection of methods, it concerns a large group of patients and presents new facts about CRB1 dystrophies.

The manuscript is one of the most complete and properly prepared among those that I have been able to review in the last year. Retinal dystrophies are not as hot a topic as AMD or DME in retinology, so readers' interest is more limited, but this work deserves a lot of praise.

It is correctly and interestingly written and also well illustrated. I am convinced that it will be cited many times by both retinologists and geneticists.

Author Response

Please find attached our revised manuscript entitled CRB1-related retinal dystrophies in a cohort of 50 patients: a reappraisal in the light of specific Müller cell and photoreceptor CRB1-isoforms.” Firstly, we would also like to thank reviewer 2 for his highly positive comments